



# Methodology for deriving the telescope focus function and its uncertainty for a heterodyne pulsed Doppler lidar

Pyry Pentikäinen[1], Ewan James O'Connor[2,3], Antti Juhani Manninen[2], and Pablo Ortiz-Amezcua[4,5]

[1]Institute for Atmospheric and Earth System Research / Physics, Faculty of Science, University of Helsinki, Helsinki, Finland
[2]Finnish Meteorological Institute, Helsinki, Finland
[3]Department of Meteorology, University of Reading, United Kingdom
[4]Andalusian Institute for Earth System Research (IISTA-CEAMA), 18006, Granada, Spain
[5]Department of Applied Physics, University of Granada, 18071 Granada, Spain

**Correspondence:** Pyry Pentikäinen (pyry.pentikainen@helsinki.fi)

**Abstract.** Doppler lidars provide two measured parameters, radial velocity and signal-to-noise ratio, from which winds and turbulent properties are routinely derived. Attenuated backscatter, which gives quantitative information on aerosols, clouds, and precipitation in the atmosphere, can be used in conjunction with the winds and turbulent properties to create a sophisticated classification of the state of the atmospheric boundary layer. Calculating attenuated backscatter from the signal-to-noise ratio

requires accurate knowledge of the telescope focus function, which is usually unavailable. Inaccurate assumptions of the telescope focus function can significantly deform attenuated backscatter profiles, even if the instrument is focused at infinity. Here, we present a methodology for deriving the telescope focus function using a co-located ceilometer for Halo Photonics Streamline and XR pulsed heterodyne Doppler lidars. The method derives two parameters of the telescope focus function, the effective beam diameter and the effective focal length of the telescope. Additionally, the method provides uncertainty estimates

for the retrieved attenuated backscatter profile arising from uncertainties in deriving the telescope function, together with standard measurement uncertainties from the signal-to-noise ratio. The method is best suited for locations where the absolute difference in aerosol extinction at the ceilometer and Doppler lidar wavelengths is small.

## 1 Introduction

Coherent Doppler lidar systems are capable of providing accurate radial Doppler velocities at high temporal and spatial res-

olution, and have been employed across a wide range of scientific and operational fields. Meteorological applications include the retrieval of turbulent properties to determine the strength, location and source of mixing in the atmospheric boundary layer, and, with many systems having scanning capability, the retrieval of winds. Information on the targets responsible for the radial Doppler velocities measured by the Doppler lidar (e.g. aerosol, cloud, precipitation), would greatly aid the interpretation of both the velocities and the products derived from them, but this requires quantitative use of the signal power received by the

instrument.

The performance of a Doppler lidar depends on the signal-to-noise ratio, SNR, of the system, as SNR determines the radial velocity uncertainty (Rye and Hardesty, 1993; Pearson et al., 2009). The outgoing laser beam can be focused to improve the



SNR at ranges close to the focal length (Pearson et al., 2002), and this is often used to improve the Doppler lidar velocity data quality and data availability, particularly in the atmospheric boundary layer. The optimal choice of focus will depend on the atmospheric conditions at the deployment location (Hirsikko et al., 2014).

Knowledge of how the choice of instrument parameters, such as the effective focal length of the telescope, impact the SNR profile is necessary in order to obtain profiles of attenuated backscatter coefficient (Zhao et al., 1990). A comprehensive overview of the theoretical considerations in determining the performance of coherent Doppler lidar systems was given by Frehlich and Kavaya (1991), who provided analytical expressions for deriving the expected signal measured by the coherent detector for a given target for a range of instrument configurations, including analytical expressions for the telescope focus function (also termed coherent responsivity). Most analytical expressions assume ideal Gaussian beams, which may not always be appropriate (Hill, 2018), hence experimental approaches have also been used to determine the impact of beam aberrations (Hu et al., 2013).

The profile of attenuated backscatter coefficient has the potential to be used in real time by weather forecasters (Illingworth et al., 2019), as it can be used in the same manner as for ceilometers. This includes the detection of liquid, supercooled-liquid, mixed-phase and ice clouds (Hogan et al., 2003; Van Tricht et al., 2014; Tonttila et al., 2015), aerosol layer and mixing-height determination (Flentje et al., 2010; Kotthaus and Grimmond, 2018), and retrieving precipitation parameters (Lolli et al., 2018).

In addition to providing velocity estimates for wind and turbulence, the inclusion of the profile of attenuated backscatter coefficient is advantageous for Doppler lidar boundary layer classification schemes (Tucker et al., 2009; Harvey et al., 2013; Manninen et al., 2018) by enhancing the discrimination between aerosol, cloud and precipitation, and can be used for tracking elevated aerosol plumes (Hannon et al., 1999). The combination of attenuated backscatter profiles from coherent Doppler lidars with other profiling instruments permits additional retrievals; for example, together with a ceilometer (Westbrook et al., 2010b), or with a cloud radar (Träumner et al., 2010), can yield drizzle drop size and precipitation rate.

Therefore, an accurate profile of attenuated backscatter coefficient requires confidence in the parameters used to generate the telescope focus function. The parameters may not be known a priori, or may differ from what is assumed, and incorrect values can result in artefacts and very large biases in attenuated backscatter coefficient. We present a methodology for deriving the parameters of the telescope focus function experimentally from co-located Doppler lidar and ceilometer observations, together with the uncertainties in the function parameters. The ceilometer, for which the overlap function is known, provides our reference attenuated backscatter profiles. This methodology is relevant for coherent Doppler lidars designed for meteorological applications with maximum ranges suitable for observing the full extent of the boundary layer and beyond. Note that a calibration constant may still need to be determined and applied after implementing the calculated telescope focus function to retrieve the profile of attenuated backscatter coefficient (Westbrook et al., 2010a; Chouza et al., 2015).

The theoretical description of the telescope focus function is outlined in Sec. 2. In Sec. 3, we introduce the instruments and the methodology for deriving the parameters of the telescope focus function experimentally. An iterative least-squares regression using weighted-Mean-Square-Error (MSE) is used to find the best solution for the telescope focus function, where the weights represent the measurement uncertainties in both instruments. The use of long time periods (one year or more) also provides an estimate of the uncertainties in the parameters for the telescope focus function, which can then be propagated





through to uncertainties in the retrieved attenuated backscatter coefficients. The methodology is applied to different instruments in multiple locations in Sec. 4 and the validation of the method is presented in Sec. 5.

## 2 Theory

### 2.1 Telescope focus function

Following Frehlich and Kavaya (1991), the coherent Doppler lidar equation can be expressed as

$$\mathrm{SNR}(R) = \frac{\eta c E}{2 h \nu B} \frac{A_e(R)}{R^2} \beta'(R), \tag{1}$$

where SNR is the signal-to-noise ratio, varying as a function of range, $R$, from the instrument, $\beta'$ is the attenuated backscatter coefficient, $c$ is the speed of light, $E$ is the beam energy, $h$ is Planck's constant, $\nu$ is the optical frequency, $B$ is the receiver bandwidth, and $A_e$ is the effective receiver area.

For a monostatic system emitting a circular Gaussian beam, using a circular aperture, and having matched filters, the effective receiver area is given by (Frehlich and Kavaya, 1991; Henderson et al., 2005)

$$A_e(R) = \frac{\pi D^2}{4 \left(1 + \left(\frac{\pi D^2}{4 \lambda R}\right)^2 \left(1 - \frac{R}{f}\right)^2 + \left(\frac{D}{2 \rho_0}\right)^2\right)}, \tag{2}$$

where $D$ is the $1/e^2$ effective diameter of a Gaussian beam, $\lambda$ is the laser wavelength, $f$ is the effective focal length of the telescope for the transmitter and receiver, and $\rho_0$ is a turbulent parameter, also termed transverse field coherence length.

Collecting the range-dependent terms, we obtain a unitless telescope focus function

$$T_f(R) = \frac{A_e(R)}{R^2}, \tag{3}$$

which is also termed the coherent responsivity (Frehlich and Kavaya, 1991).

The profile of attenuated backscatter coefficient is then obtained by rearranging (1)

$$\beta'(R) = \frac{2 h \nu B}{\eta c E} \frac{\mathrm{SNR}(R)}{T_f(R)}. \tag{4}$$

Figure 1a shows how $T_f(R)$ depends on the telescope focal length, $f$, and Fig. 1b how $T_f(R)$ depends on $D$. Both figures show that the apparent focus — i.e range to the $T_f(R)$ maximum — is always closer than $f$, and that decreasing $D$ shortens the apparent focus. This makes estimation of the parameters by eye in $T_f(R)$ prone to errors, since the apparent focus cannot be translated into $f$ without knowledge of $D$.

Figure 1c shows that even if the telescope is focused at infinity, knowledge of the $D$ is essential to derive attenuated backscat-
ter coefficient profiles. While the gradient of $T_f(R)$ may be independent of $D$ at the near and far ranges, the relative magnitude is not, and the potential variation is high in the range of the profile that is commonly of most interest.





**Figure 1.** Telescope focus functions for: a) varying $f$ with $D$=70 mm, b) varying $D$ with $f$=1000 m, c) varying $D$ with $f$ being infinity.



## 2.2 Uncertainty in attenuated backscatter coefficient

Assuming that the parameters $T_f(R)$ and SNR are independent, and have uncertainties that can be described as Gaussian, the random uncertainty in attenuated backscatter coefficient is

$$\sigma_{\beta'} = \sqrt{\sigma_S{}^2 + \sigma_{T_f}{}^2}, \tag{5}$$

where $\sigma_S$ is the uncertainty in the Doppler lidar SNR, and $\sigma_{T_f}$ is the uncertainty in $T_f(R)$. An expression for deriving $\sigma_S$ is given by Manninen et al. (2018), and we describe our method for obtaining $\sigma_{T_f}$ in Section 4.2.

## 3 Application to data

There are 3 range-dependent unknowns in (2): $f$, $D$, and $\rho_0$. We assume that we can neglect $\rho_0$, and describe a method for estimating $f$ and $D$, together with their uncertainties, which can then be propagated to obtain the uncertainty in the attenuated

backscatter coefficient.

### 3.1 Instruments

We used measurements taken from the U.S. Department of Energy Atmospheric Radiation Measurement (ARM, Mather et al., 2016) observatories. We selected 5 sites with co-located ceilometer and Doppler lidar instruments: Southern Great Plains, US (SGP); Tropical West Pacific, Darwin, Australia (Darwin); Barrow, Alaska, US (NSA); Graciosa, Azores (Graciosa); Ascension

Island, Atlantic, UK (Ascension).

     The Doppler lidars operated by ARM comprise both Halo Photonics Streamline, and Streamline XR versions. These are commercially available heterodyne pulsed systems capable of full-hemispheric scanning and operated at a temporal resolution of 1-2 s (see Table 1). The focus for the Streamline version can be set by the operator, whereas the Streamline XR has the focus set by the manufacturer; however ARM has had some instruments upgraded from their original specification.

The ceilometer at all sites was a Vaisala CL31 ceilometer, which has a coaxial design and full overlap before 100 m and a temporal resolution of 30 s (more specifications given in Table 2).

### 3.2 Methodology

#### 3.2.1 Telescope focus function parameter estimation

The methodology for deriving the parameters of the telescope focus function compares profiles from co-located Doppler lidar

and ceilometer using an iterative least-squares regression to find the best solution. The method follows the process diagram given in Fig. 2.

     Before input, the Doppler lidar SNR data had a background correction applied to reduce bias (Manninen et al., 2016), and data below a minimum SNR threshold of -22.2 dB was discarded. Then, both ceilometer and Doppler lidar data were averaged to a common 30-minute, 30 m vertical resolution grid, using interpolation where necessary (only for one period from Darwin).





**Table 1.** Halo Photonics Streamline and Streamline XR heterodyne Doppler lidar specifications. Values in parentheses refer to the specification of the Doppler lidar during the first period in Darwin.

| | |
|---|---|
| Wavelength | 1.5 $\mu$m |
| Pulse repetition rate | 15 kHz |
| Nyquist velocity | 19.8 m s$^{-1}$ |
| Sampling frequency | 50 MHz |
| Points per range gate | 10 (16) |
| Range resolution | 30 m (48 m) |
| Pulse duration | 0.2 $\mu$s |
| Divergence | 33 $\mu$rad |
| Antenna | monostatic optic-fibre coupled |

**Table 2.** Vaisala CL31 ceilometer specifications.

| | |
|---|---|
| Wavelength | 910 nm |
| Pulse repetition rate | 5.57 kHz |
| Range resolution | 30 m |
| Lens diameter | 14.5 cm |
| Divergence | 0.75 mrad |

The data was then filtered to select only those portions of the profiles that are considered reliable for comparison. Ceilometer data below 195 m was discarded to ensure that only data with full overlap was used.

Due to the wavelength difference between the Doppler lidar and the ceilometer, it cannot be assumed that the atmospheric backscattering properties are the same at both wavelengths. However, we are only interested in the profile shape, not the absolute values, so profiles from the Doppler lidar and the ceilometer can be compared as long as they contain only one type of scatterer, and one which can be assumed to be distributed homogeneously throughout the portion of the profile used for comparison. Hence, the portion of a profile selected for comparison should contain only one aerosol layer, no clouds, and no precipitating hydrometeors.

We removed clouds by identifying the range gate 150 m below the cloud base detected by the ceilometer and excluding all data beyond this. Elevated aerosol layers and precipitating hydrometeors were filtered out by identifying layers using a convolution of the ceilometer profile with a haar-wavelet to detect changes in the gradient. The base of the second layer





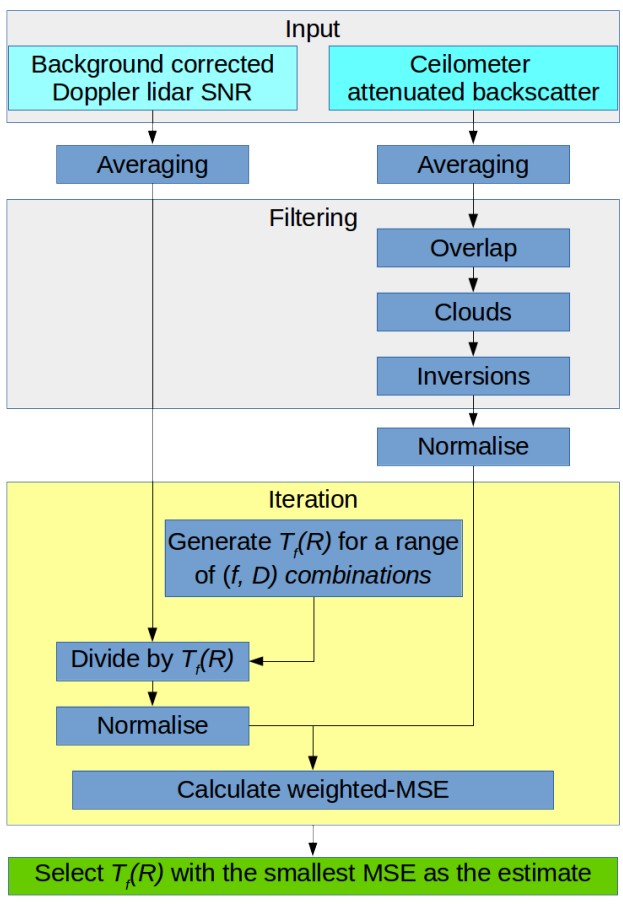

**Figure 2.** Process diagram of the telescope focus function parameter estimation.

was identified where the gradient was increasing over 2 range gates and all data above this was discarded. This process also eliminates noisy profiles with low SNR.

The $T_f(R)$ parameter estimation is performed on a profile-by-profile basis for each profile where the filtering process leaves 8 successive range gates present. From equation 4, dividing the Doppler lidar SNR profile with the appropriate $T_f(R)$ will

5 generate a Doppler lidar attenuated backscatter profile whose shape should match that of the ceilometer attenuated backscatter profile.

We use a brute-force approach to iterate through a range of reasonable $f$ and $D$ values, generating a corresponding $T_f(R)$ and Doppler lidar attenuated backscatter profile for each combination of values. The ceilometer profile and resulting Doppler lidar attenuated backscatter profiles are normalised so that the integral value of the unfiltered portion is unity. We then use

10 a least-squares regression using weighted-MSE to find the best solution (smallest MSE), where the weights represent the



measurement uncertainties in both instruments. Collecting results over many profiles results in a bi-variate distribution; the peak of this distribution is chosen as the best estimate of $f$ and $D$, and hence the best estimate of $T_f(R)$ using (3).

### 3.2.2 Outlier removal

Occasionally, data of poor quality passes the filtering step in Fig. 2. The most common issues are noisy ceilometer data, and a bias in the Doppler lidar SNR profiles. If not screened, these occasional profiles result in significantly altered $T_f(R)$ estimation. Any noise in the ceilometer data is magnified by the profile length often being relatively short, and hence large uncertainty in even a single range gate can skew the regression. Doppler lidar SNR bias will impact the normalisation process, changing the $T_f(R)$ selected by the method due to the now incorrect profile shape. Due to the non-linearity of the $T_f(R)$ parameter estimation process, these issues result in regression solutions wildly inconsistent with the estimates based on good data. These outliers, which do not fall within the normal uncertainty observed in good data, are then removed from the bi-variate distribution of solutions before calculating the uncertainty estimates.

We used the median absolute deviation, MAD (Huber and Ronchetti, 2009; Leys et al., 2013), to distinguish outliers in the bivariate distribution of estimated $f$ and $D$. MAD can be calculated using

$$MAD = b \, med\{|x_i - med\{x_i\}|\},  \tag{6}$$

where $b = 1.4826$ when the distribution excluding the outliers is normal. However, the distribution of $f$ and $D$ may not meet this criterion due to the non-linearity of $T_f(R)$ and the computational $T_f(R)$ estimation process. We expect the distributions of $D$ and $f^{-2}$ to be close to normal and will use $f^{-2}$ rather than $f$ to determine outliers. Additionally, the peak of the bi-variate distribution may not always coincide with the medians of the uni-variate $D$ and $f^{-2}$ distributions, and, hence, we use a modified form of (6),

$$MAD = b \, med\{|x_i - peak\{x_i, y_i\}|\}.  \tag{7}$$

We selected 3 MADs as the threshold for flagging outliers:

$$\sqrt{\left(\frac{f_i^{-2} - med\{f_i^{-2}\}}{MAD_{f^{-2}}}\right)^2 + \left(\frac{D_i - med\{D_i\}}{MAD_D}\right)^2} \geq 3.  \tag{8}$$

Assuming the distribution excluding the outliers to be normal, 3 MADs correspond to 3 standard deviations of the underlying distribution. In cases where $f$ is at infinity, all estimates with a finite $f$ will be flagged as outliers.

## 4 Results

### 4.1 Parameter estimation

We applied the $T_f(R)$ estimation method to Doppler lidars at 5 ARM atmospheric observatories. Figure 3a shows the distribution of $f$ and $D$ calculated for the Doppler lidar operating at Darwin in northern Australia between 21 June 2011 and 22

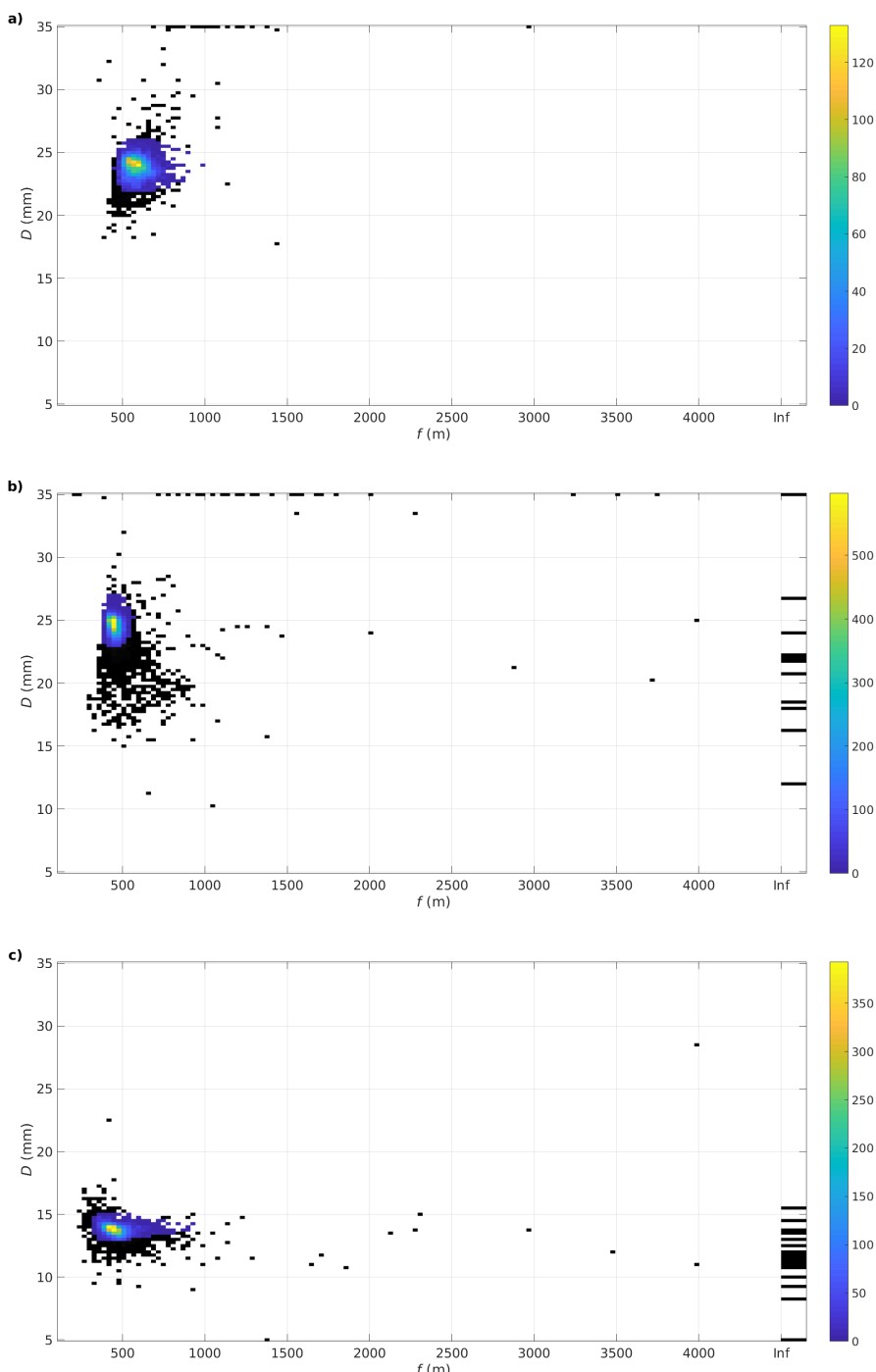

**Figure 3.** Distributions of $T_f(R)$ parameter estimates from a) Darwin 21 June 2011 to 22 July 2012, b) SGP 1 January 2015 to 2 May 2016, c) SGP 3 May 2016 to 5 June 2017. Outliers filtered using MAD $\geq 3$ are marked in black.





**Table 3.** Best estimates of $f$ and $D$ together with their uncertainties for Doppler lidars at 5 ARM sites.

| Location | Period | $f$ | $D$ | Available Profiles | Total estimates | Good estimates |
|---|---|---|---|---|---|---|
| Ascension | 20160906–20170930 | 550±34 m | 25.3±0.5 mm | 18586 | 62 | 56 |
| Darwin | 20110621–20120722 | 590±62 m | 24.0±0.7 mm | 18988 | 3684 | 3528 |
| Darwin | 20120921–20140626 | 545±53 m | 25.0±0.8 mm | 30836 | 5046 | 4878 |
| Graciosa | 20150124–20161114 | 625±80 m | 23.5±0.7 mm | 31124 | 3737 | 3161 |
| NSA | 20140730–20171231 | Infinity | 11.8±1.5 mm | 56832 | 1132 | 589 |
| SGP | 20150101–20160502 | 440±29 m | 25.0±0.7 mm | 22916 | 9198 | 8426 |
| SGP | 20160503–20170605 | 425±74 m | 14.0±0.4 mm | 14212 | 5814 | 5337 |

July 2012. This Doppler lidar is a Streamline and the distribution of $f$ is positively skewed, as explained in section 3.2.2. The distribution displays a slightly wider peak than expected for a normal distribution.

Figure 3b shows the distribution of $f$ and $D$ for the Streamline Doppler lidar operating at SGP from 1 January 2015 to 2 May 2016. The distribution close to the peak is really tight, while the outliers have substantial spread. Many of the poor estimates

responsible for the outlier spread occur during January and February in both years, while for the rest of the period the estimates are remarkably consistent. On 3 May 2016 the Doppler lidar at SGP was changed to a Streamline XR and Fig. 3c shows the distribution of $f$ and $D$ from 3 May 2016 to 5 June 2017. The change in instrument version, from Streamline to Streamline XR is clearly seen in the change in $D$, whereas the best estimate for $f$ did not change. However, inspecting the data by eye would suggest a significantly shorter apparent focus, and the $T_f(R)$ calculated using the best estimates for $f$ and $D$ also exhibits a

significantly shorter apparent focus. Consequently, the Streamline XR in SGP has been noted to suffer from poor SNR at the boundary layer top.

The bi-variate distributions of $f$ and $D$ show notable variations in how tight they are around the peak, and is likely a result of differences in data quality between the instruments. The best estimates of $f$ and $D$ and their uncertainties for all sites are presented in Table 3. The Doppler lidar measurements at Darwin were split into two periods, as there was a two month break in

the measurements between these two periods. We performed the $T_f(R)$ parameter estimation separately for both periods. The best estimates from these periods differ from each other, which is expected as some adjustments were made to the instrument. The telescope focal length for this instrument is directly adjustable by any operator while the beam diameter is set by the manufacturer and is not modifiable by the operator. We note that the $D$ estimates are the same for these two periods within the margin of error calculated.

For the sites and instruments selected here, only the Doppler lidar at NSA had $f$ set to infinity. In fact, all Streamlines have $D$ in the vicinity of 25 mm, whereas $D$ for the Streamline XR versions is about half this. Nevertheless, the variation between instruments of the same version is not negligible and should be taken into account when calculating $T_f(R)$ and then attenuated backscatter.





The final step to obtain attenuated backscatter profiles is to apply a calibration constant, which can be achieved using the liquid cloud calibration method (Westbrook et al., 2010a; O'Connor et al., 2004).

The parameters $f$ and $D$ calculated for period 1 in Darwin have been used to derive $T_f(R)$ and the results applied in Fig. 4. This shows the utility of the method, able to provide reliable Doppler lidar attenuated backscatter profiles in Fig. 4b that show

no over correction below 1 km and display similar in-cloud values to the ceilometer in Fig. 4c. It is expected that the aerosol attenuated backscatter coefficients will differ due to the different scattering properties of aerosol at the different wavelengths; the scattering properties of cloud droplets remain similar at the two wavelengths (O'Connor et al., 2004; Westbrook et al., 2010a).

## 4.2    Uncertainty

A computational method was used to calculate the uncertainty in the estimated $T_f(R)$ as it is a non-linear function of $f$ and $D$. We used Monte Carlo simulation (MCS) (Morgan and Henrion, 1990) where a distribution of input values is fed into a model, here the effective receiver area equation (2), and the uncertainty is obtained from the distribution of the output. The input values can be created either from observed statistics, or by bootstrapping, i.e. re-sampling the data. We created three different sets of input values for our MCS:

1. Re-sampling the individual estimates of $f$ and $D$ provided directly by the $T_f(R)$ estimation method (i.e. those displayed in Fig. 3) after excluding outliers.

2. Generating the values from the statistics presented in Table 3, assuming that $D$ and $f^{-2}$ are normally distributed and independent, $N(f^{-2}, D)$.

3. Generating the values from statistics presented in Table 3, assuming $D$ and $f$ to be normally distributed and independent,

$N(f, D)$.

For each set of input values, the relative uncertainty is expressed in terms of the standard deviation of the distribution of MCS-simulated telescope focus functions, $T_\phi(R)$,

$$\sigma_{T_f}(R) = \frac{\sigma_{T_\phi}(R)}{T_f(R)}, \tag{9}$$

noting that the obtained uncertainty is range-dependent.

Examples of the three input parameter distributions are presented in figure 5. Re-sampling (Fig. 5a) is the most accurate method as it does not require assumptions about the parameter distributions and their independence. We recommend re-sampling as the primary method for uncertainty calculation. Using the $N(f^{-2}, D)$ distribution (Fig. 5b) produces a set of input values that appear to be a reasonable approximation, except that the distribution is not as tight around the peak. Using the $N(f, D)$ distribution (Fig. 5c) produces a set of input values that tend to over-emphasise shorter values of $f$, and under-

emphasise higher values. We also note that the central bin of the re-sampled distribution contains 50% more samples than the central bin of the statistically-generated distributions do. We presume that this is a consequence of the variation in SNR not being necessarily normally distributed.

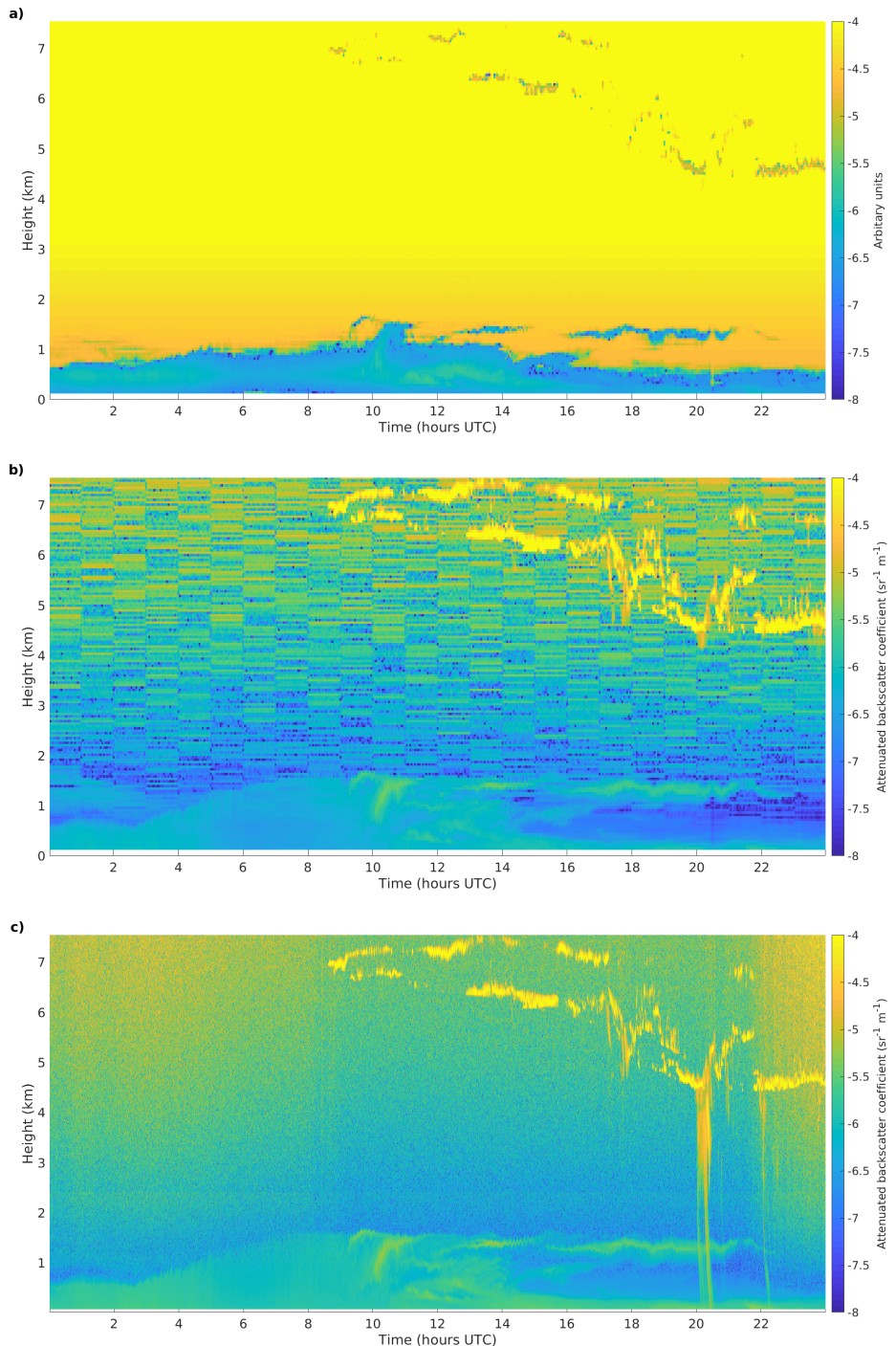

**Figure 4.** a) Doppler lidar attenuated backscatter coefficient assuming a generic $T_f(R)$, b) corrected Doppler lidar attenuated backscatter coefficient, c) ceilometer attenuated backscatter coefficient, from Darwin on 28 May 2012.





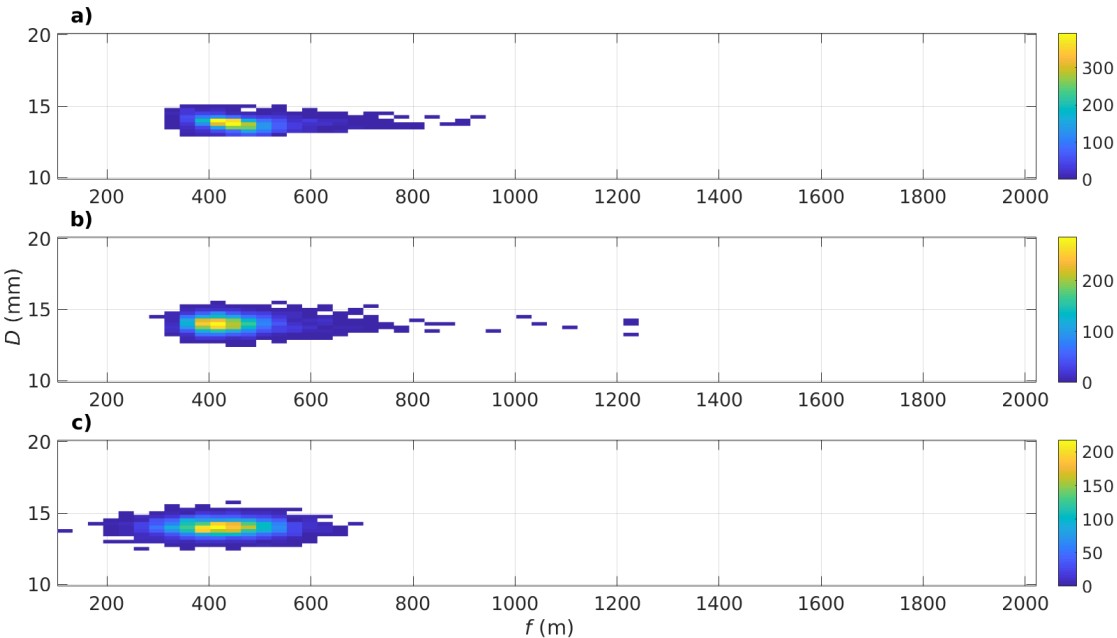

**Figure 5.** Distributions of the MCS input values used for calculating $\sigma_{T_f}(R)$. Values are obtained from a) re-sampling, b) assuming $N(f^{-2}, D)$, c) assuming $N(f, D)$. All distributions contain 5337 samples.

**Table 4.** $\sigma_{T_f}(R)$ uncertainty envelopes generated using MCS with three different sets of input values.

| Location | Period | Re-sampling | $N(f^{-2}, b)$ | $N(f, b)$ |
|---|---|---|---|---|
| Ascension | 20160906–20170930 | 0.14 | 0.14 | 0.16 |
| Darwin | 20110621–20120722 | 0.20 | 0.22 | 0.25 |
| Darwin | 20120921–20140626 | 0.22 | 0.21 | 0.27 |
| Graciosa | 20150124–20161114 | 0.21 | 0.19 | 0.29 |
| NSA | 20140730–20171231 | 0.32 | 0.31 | 0.30 |
| SGP | 20150101–20160502 | 0.18 | 0.19 | 0.21 |
| SGP | 20160503–20170605 | 0.12 | 0.12 | 0.23 |

Figure 6a displays $\sigma_{T_f}(R)$ for Darwin showing the range-dependence of the uncertainty, with much larger uncertainties for ranges close to either side of the focus ($f = 590\ m$). The profile of uncertainties obtained with each set of MCS input values exhibit a similar shape, with $N(f^{-2}, D)$ being closer to re-sampling than $N(f, D)$ in the near field.





**Figure 6.** Relative telescope focus function uncertainties, $\sigma_{T_f}(R)$, generated using MCS with three different sets of input values for a) Darwin 21 June 2011 to 22 July 2012, and b) NSA 30 July 2014 to 31 December 2017




Figure 6b displays $\sigma_{T_f}(R)$ for the Doppler lidar at NSA which has $f$ set to infinity, therefore $\sigma_{T_f}(R)$ is only dependent on the uncertainty in $D$. Note the reduced uncertainties around 200-400 m, which are expected when examining Fig. 1c.

The largest value of $\sigma_{T_f}(R)$ provides the uncertainty envelope value for each site, which is presented in Table 4. Re-sampling provides values ranging from 0.12 for the updated instrument at SGP, to 0.32 at NSA. MCS values created using $N(f^{-2}, D)$
provided similar values, whereas MCS using $N(f, D)$ often provided much larger uncertainties.

## 5    Validation

The liquid cloud calibration method (O'Connor et al., 2004; Westbrook et al., 2010a) is used to determine a calibration constant by integrating attenuated backscatter profiles containing fully attenuating liquid clouds, which have well-constrained apparent lidar ratio, $\eta S$, where $\eta$ is a multiple scattering factor and $S$ is the lidar ratio. In the absence of multiple scattering, $\eta S$ can be
assumed to be independent of the height of the cloud.

This calibration method can be used to evaluate the estimated $T_f(R)$ for Doppler lidar by checking whether the attenuated backscatter profiles obtained for the Doppler lidar after applying $T_f(R)$ indeed provide similar $\eta S$ values for liquid clouds at different heights.

Figure 7 shows examples of Doppler lidar attenuated backscatter profiles after calibration and the derived apparent lidar ratio
at two sites, Darwin and NSA. These sites have different values of $f$, Darwin has $f = 590$ m and NSA has $f$ set to infinity. For both cases, liquid clouds are present throughout the day with altitudes varying from 2 to 6 km. When fully attenuating liquid clouds are present, the apparent lidar ratio is close to the expected value of 20 sr, regardless of the height of the cloud, thus confirming that the method of estimating $T_f(R)$ is valid.

### 5.1    Limitations

Table 3 shows that the proportion of data that can be used for the $T_f(R)$ parameter estimation varies significantly from site to site. Over a third of the available profiles from SGP are used, whereas only 0.3% pass the filtering for Ascension. The lack of suitable profiles at Ascension is explained by the almost constant low cloud cover at this site, with very few profiles having a sufficient number of successive range gates.

Data quality is also a limiting factor so at sites with very low aerosol optical depth, AOD, such as NSA, the Doppler lidar
SNR decreases so rapidly that again there are few profiles having a sufficient number of successive range gates. Low AOD also impacts the performance of the ceilometer, with 48% of the estimates at NSA discarded as outliers even after the initial filtering was performed. While the outlier removal can separate the good and the poor estimates, the largest uncertainty in $D$ was at NSA. We attempted to perform the $T_f(R)$ parameter estimation on Doppler lidar from an ARM campaign in Cape Cod, but could not obtain reliable estimates due to the low SNR of the ceilometer data.
The $T_f(R)$ parameter estimation method is suitable only in situations where there is minimal difference in atmospheric extinction within the aerosol layer between the two instrument wavelengths of 910 nm and 1500 nm. Using AERONET AOD measurements collocated at the ARM atmospheric observatories, the median difference in AOD at 870 nm and 1640 nm varied



**Figure 7.** Doppler lidar attenuated backscatter coefficient and apparent lidar ratio, $\eta S$, from a) Darwin on 8 May 2012, b) NSA on 20 August 2014.





from 0.016 and 0.027, which should correspond closely to what might be expected for the difference between ceilometer and Doppler lidar. Very occasional periods of notable AOD differences were observed at some sites, but including these profiles in timeseries extending beyond a year will have negligible impact on the $T_f(R)$ parameter best estimate. However, there were breaks in the AOD measurements, and some periods experiencing a significant differential extinction may have gone unnoticed.

An additional filter using AERONET AOD measurements to remove profiles experiencing significant differential extinction could be included in Fig. 2 for those sites where this may be an issue.

## 6 Conclusions

We have developed a method for deriving the telescope focus function and its uncertainty for Halo Photonics Streamline and XR Doppler lidars. The method compares profiles of the Doppler lidar SNR to profiles of attenuated backscatter coefficient

from a collocated ceilometer, producing estimates for two parameters of the $T_f(R)$; the effective focal length for the telescope, $f$, and the $1/e^2$ effective diameter of a Gaussian beam, $D$. This method was developed because it also provides uncertainties in $f$, $D$ and $T_f(R)$, necessary for quantitative use of the subsequently derived attenuated backscatter profiles. The method can be used to check the manufacturer specifications for these parameters, calculate them if not known, and also check their stability over time.

The method was applied to data from Doppler lidars with different configurations deployed at 5 ARM sites. Relative uncertainties in $f$ for these instruments ranged from 6% to 17% with the median uncertainty being 10%; the relative uncertainty in $D$ ranged from 2% to 12% with median of 3%. The uncertainty in $T_f(R)$ was calculated using Monte Carlo simulation, using 3 methods to prepare the input values. We recommend the direct re-sampling method, but reasonable results were obtained used statistically-derived input values assuming a normal distribution. The envelope of relative uncertainties in $T_f(R)$ ranged

from 12% to 32%, and depend on both the instrument configuration and the instrument location. We also show that, even for a Doppler lidar with the focus set at infinity, uncertainty remains in estimating $T_f(R)$ arising from the uncertainty in $D$. The method was validated by calculating the apparent lidar ratio of fully attenuating liquid clouds from $T_f(R)$ corrected profiles of Doppler lidar attenuated backscatter.

The $T_f(R)$ estimation method is suitable only for conditions where the differential extinction at the two wavelengths of

the Doppler lidar and the ceilometer is small, which can be identified, for example, using AOD from co-located AERONET observations.

*Competing interests.* The authors declare that they have no conflict of interest.

*Acknowledgements.* This study was supported by the U.S. Department of Energy's Atmospheric System Research (ASR), an Office of Science, Office of Biological and Environmental Research (BER) program, under Contract DE-SC0017338. The Doppler lidar and ceilometer





data used in this study were obtained from the Atmospheric Radiation Measurement (ARM) user facility, managed by the Office of Biological and Environmental Research for the U.S. Department of Energy Office of Science.





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
