# Peer review of "Methodology for deriving the telescope focus function and its uncertainty for a heterodyne pulsed Doppler lidar"

_Atmospheric Measurement Techniques, 2019_

## Referee Comment (RC1) · Anonymous Referee #1 · 30 Jan 2020

The manuscript deals with a methodology that can be used to derive the telescopic functions of a pulsed Doppler lidar. The idea is to use the information on the lidars telescopic functions to derive the attenuated backscatter profile from the SNR signal from the wind- lidar. The telescopic functions are estimated by comparing (by iteration) with the attenuated backscatter profile measured by a ceilometer.

After having read the paper several times I am still in doubt whether the methodology is intended for applied use or if is a purely academic exercise. I would like the authors to put more emphasis on the use of wind-lidars in practical applications for the measurements of attenuated backscatter profiles and what can achieved by such

measurements from wind-lidars.

1) It needs to be clarified why the data filtering is so strong, why are there so few good profiles out of so many available profiles in table 3. Are some of these data simply considered outliers - it is always dangerous to neglect outliers.

2) How much does the improved telescopic functions improve the attenuated backscatter profile as compared to the information from the factory setting of the telescopic functions?

3) How well does the attenuated backscatter profiles determined from the wind lidar SNR profile compare to the profiles observed by ceilometer. Only a few examples are shown in the paper, and a real quantification based on many (all) profiles from these rich data sets would be an considerable improvement to the paper. The main question is if the wind lidar is able on a routine basis to produce reliable profiles of attenuated backscatter profiles. A ceilometer is a very cheap instrument compared to a wind lidar, is it still recommendable to have a ceilometer next to a wind lidar or can the ceilometer be omitted and the backscatter profile determined with sufficient accuracy from the SNR?

Minor remarks

1) Line 28 – page 5. Why is the threshold chosen to be 22.2 dB, the number sounds arbitrary. Why not simply set a very high threshold value for this exercise – e. g. -15 dB, to secure high quality data?

2) Line 28 page 5, If observations below -22.2 dB are discarded, the averaged SNR will be biased – is this accounted for?

3) Line 29 page 5. Explain what is meant by "using interpolation where necessary".

4) Line 9 page 6. How is the cloud base detected? Do you use a threshold method (if yes what is the threshold) or a more sophisticated method?

5) Line 17, Page 8: Explain why you expect f-2 to be superior.

6) Why do you mix two parameters for the flagging in Eq. (8), It seems more natural to flag the individual parameter.
* * *

---

## Referee Comment (RC2) · Anonymous Referee #2 · 14 Feb 2020

**Comments on the article "Methodology for deriving the telescope focus function and its uncertainty for a heterodyne pulsed Doppler lidar" by Pentikainen *et al.**

February 14, 2020

**General comments**

The paper presents a practical method for characterizing the "telescope aperture function" of a pulse heterodyne lidar. As explained in the article, this function is required to correct the intensity of the signals recorded by the instrument (termed $SNR$ as it is normalized to the level of white noise in the heterodyne lidar) from the variations of the instrument sensitivity with the range. This function is more complicated than with a direct-detection aerosol lidar as it is not given by the overlap between the laser beam and the telescope aperture, but has more to do with the efficiency of the heterodyne detection. However, an expression exists that predicts how it varies with the range as a function of two system parameters, namely the size of the telecope aperture $D$ and the focal length $f$. The idea is to tune these two parameters until the corrected signal intensities match the attenuated backscatter measured by a nearby ceilometer. Of course, this method requires the presence of ceilometer nearby, but ceilometers are rather cheap instruments, and are deployed in great numbers in meteorological observation networks. The method opens the possibility to use a heterodyne wind lidar as an aerosol lidar and thus combine with a unique instrument the measurement of wind and aerosol backscatter profiles. This is of great interest for the characterization of pollution transport or the study of the atmospheric boundary layer.

There are several limtations to the method. One deals with the difference of the laser wavelength of a ceilometer (usually close to $0.9\mu m$) and a pulsed heterodyne lidar ($\lambda \approx 1.5\mu m$). The value of the attenuated backscatter are different at the two wavelengths, and the difference is dependant of the nature the aerosols. The methode would thus be in trouble if several layers of different aerosols are present in the laser beam. This limitation is clealry discussed in the text. There is, however, a second limitation. The heterodyne efficiency does not depend solely of the instrument paramaters, but also on the optical turbulence. This dependcy appears in equation (2) of the article with the $\rho_0$ parameter. It is assumed in article that $\rho_0 >> D$ so that its effect on $A_e(R)$ can be neglected.

This assumption is not justified. In practice, it can happen that the turbulence significantly degrades the heteridyne efficiency of the lidar. This is particularly the case when the beam is directed horizontally, a few meters above the ground, on a hot, sunny day. In the article, the vertical (or close to vertical) direction of the beam should alleviate the degradation as the optical turbulence drops very rapildy with the altitude, but it would be worth to have a short calculation of $\rho_0$ as a function of the range using a typical profile of $C_n^2$ and the formulation of Frehlich and Kavaya for $\rho_0$ (equation 165).

It is written in the abstract that the method proposed in the article is applicable to Halo Photonics heterodyne Doppler lidars. It is clear that it is tested on data from Halo Photonics lidars, but I do not see why it could not be applicable to heterodyne lidars from other manufacturers as long as they provide measurements of $SNR$. If that is true, it would be worth mentionning it in the abstract as it widens the applicability of the method.

**Specific comments**

1. In the paragraph that follows equation (1), the meaning of the $\eta$ term is not explained.

2. Equation (5): the equation applies to relative uncertainties. This shall be made clear.

3. Line 28: why this $-22.2dB$ SNR threshold?

4. Table 4 on page 13: the meaning of $b$ in $N(f^{-2}, b)$ and $N(f, b)$ is not explained in the legend.

**Conclusion**

This is an interesting paper that proposes a useful and practical way for the characterization of the variation of the receiving efficiency of a heterodyne lidar with the range. Minor modifications would improve its quality. It deserves publication.

---

## Author Comment (AC1) · 27 Mar 2020

We have addressed all of the points raised by the reviewer (copied here and shown in black text), and include our responses to each point below (in blue text). Where there has been a major change in the manuscript we provide the original text (in black italics) and the new text (in blue italics).

**1 Anonymous Referee 1**

The manuscript deals with a methodology that can be used to derive the telescopic functions of a pulsed Doppler lidar. The idea is to use the information on the lidars telescopic functions to derive the attenuated backscatter profile from the SNR signal from the wind- lidar. The telescopic functions are estimated by comparing (by iteration) with the attenuated backscatter profile measured by a ceilometer.

After having read the paper several times I am still in doubt whether the methodology is intended for applied use or if is a purely academic exercise. I would like the authors to put more emphasis on the use of wind-lidars in practical applications for the measurements of attenuated backscatter profiles and what can achieved by such measurements from wind-lidars.

We have also included a statement outlining the advantage of obtaining attenuated backscatter and Doppler velocity measurements in the same measurement volume, with reference to the potential of deriving mass-fluxes, e.g. aerosol or cloud.

*There is also an advantage to obtaining attenuated backscatter and Doppler velocity measurements in the same measurement volume, since this will simplify the calculation of cloud or aerosol mass-fluxes (Engelmann* et al.*, 2008).*

1) It needs to be clarified why the data filtering is so strong, why are there so few good profiles out of so many available profiles in table 3. Are some of these data simply considered outliers - it is always dangerous to neglect outliers.

The majority of the filtering is to remove profiles that are not suitable for this method - profiles that contain clouds, precipitation or multiple aerosol layers (some sites are more cloudy than others). This explains the reduction of 'available profiles' to 'total estimates'. Then, the outlier filtering using MAD is the difference between 'total' and 'good estimates' and is usually $< 10\%$, except for the NSA site. The SNR for both instruments is quite low at NSA due to the low aerosol loading; hence the outlier filtering is stronger. The outliers are still plotted in Fig. 3 but not used in calculating the uncertainties.

2) How much does the improved telescopic functions improve the attenuated backscatter profile as compared to the information from the factory setting of the telescopic functions?

This will vary from instrument to instrument. $D$ is often quite close to the nominal value provided by the manufacturer, but $f$ may not be or may

be unknown, and for some models, $f$ can also be adjusted by some known or unknown amount by the operator.

3) How well does the attenuated backscatter profiles determined from the wind lidar SNR profile compare to the profiles observed by ceilometer. Only a few examples are shown in the paper, and a real quantification based on many (all) profiles from these rich data sets would be an considerable improvement to the paper. The main question is if the wind lidar is able on a routine basis to produce reliable profiles of attenuated backscatter profiles. A ceilometer is a very cheap instrument compared to a wind lidar, is it still recommendable to have a ceilometer next to a wind lidar or can the ceilometer be omitted and the backscatter profile determined with sufficient accuracy from the SNR?

The absolute values of the attenuated backscatter from the Doppler lidar and the ceilometer are not expected to be the same due to the difference in the wavelength, but for a homogeneous aerosol layer, the profile shape will match. The Doppler lidar is expected to then provide reliable profiles on a routine basis after applying the telescope focus function calculated using this method together with a calibration factor calculated using e.g. (Westbrook et al., 2010a).

Calculation of the telescope focus function can be made from a short time-series next to a ceilometer; the instrument can then be moved to another location, for campaigns for example. In practice this method can be performed during commissioning, and in principle, the manufacturer could also provide this service.

**1.1 Minor remarks**

1. Line 28 – page 5. Why is the threshold chosen to be 22.2 dB, the number sounds arbitrary. Why not simply set a very high threshold value for this exercise – e. g. -15 dB, to secure high quality data?

   The threshold is based on the SNR limit in Manninen et al.(2016) as the data we used has been processed with the same method. This threshold is based on the expected noise floor for the instruments considered here (Halo Streamline and Streamline XR) and should probably be modified for different instruments. A citation for the threshold has been added. We agree that a higher threshold could be used to secure high quality data, but we also wanted to test the applicability of the method in situations where mostly low SNR is expected, e.g. at the NSA site in Alaska where the aerosol loading is very low.

2. Line 28 page 5, If observations below -22.2 dB are discarded, the averaged SNR will be biased – is this accounted for?

   The order of these two steps was written incorrectly in the manuscript. The averaging was done prior to discarding the low SNR data, and the threshold then applied to the averaged data. The manuscript has now been corrected.

*Before input, the Doppler lidar SNR data had a background correction applied to reduce bias (Manninen et al., 2016), and data below a minimum SNR threshold of -22.2 dB was discarded. Then, both ceilometer and Doppler lidar data were averaged to a common 30-minute, 30 m vertical resolution grid, using interpolation where necessary (only for one period from Darwin).*

*Before input, the Doppler lidar SNR data had a background correction applied to reduce bias (Manninen et al., 2016). Both ceilometer and Doppler lidar data were averaged to a common 30-minute, 30 m vertical resolution grid, using linear interpolation where necessary (only for one period from Darwin). After averaging, data below a minimum threshold of -22.2 dB (Manninen et al., 2016) was discarded.*

3. Line 29 page 5. Explain what is meant by "using interpolation where necessary".

   Interpolation may necessary to get the ceilometer and Doppler lidar data on the same vertical grid. Usually, the vertical resolution of the ceilometer data is high enough (10 m) so that 2 or 3 range gates in the vertical can be summed to match the Doppler lidar vertical resolution, however at one site the difference between the vertical resolutions of the two instruments required linear interpolation between ceilometer range gates to match the Doppler lidar resolution.

4. Line 9 page 6. How is the cloud base detected? Do you use a threshold method (if yes what is the threshold) or a more sophisticated method?

   Here we use the Vaisala cloud base detection, which uses a gradient method on the ceilometer attenuated backscatter profile. A more sophisticated shape method (e.g. Tuononen et al., 2019) could also be used but we are not so interested in the precise cloud base value, more whether a cloud layer exists in the profile - this is also why we only use data more than 150 m below cloud base.

5. Line 17, Page 8: Explain why you expect f-2 to be superior.

   When examining Equation (2), we expect f-2 to be superior to f, and f-2 is closer to the distribution observed for the telescope focus in figure 3. Additionally, figure 1a, which is plotted with range in logarithmic units also suggests this relationship.

6. Why do you mix two parameters for the flagging in Eq. (8), It seems more natural to flag the individual parameter.

   Our best estimate for the Telescope Focus Function parameters is the peak of the bi-variate (f,D) distribution, and thus for the flagging we use distance from the peak of the bi-variate distribution.

**References**

M. Tuononen, E. J. O'Connor, and V. A. Sinclair: Evaluating solar radiation forecast uncertainty, *Atmos. Chem. Phys.*, **19**, 1985–2000, doi:10.5194/acp-19-1985-2019, 2019.

Engelmann, R., Wandinger, U., Ansmann, A., Müller, D., Žeromskis, E., Althausen, D., and Wehner, B.: Lidar Observations of the Vertical Aerosol Flux in the Planetary Boundary Layer, Journal of Atmospheric and Oceanic Technology, 25, 1296–1306, doi:10.1175/2007JTECHA967.1, 2008.

---

## Author Comment (AC2) · 27 Mar 2020

We have addressed all of the points raised by the reviewer (copied here and shown in black text), and include our responses to each point below (in blue text). Where there has been a major change in the manuscript we provide the original text (in black italics) and the new text (in blue italics).

**1 Anonymous Referee 2**

**1.1 General comments**

The paper presents a practical method for characterizing the "telescope aperture function" of a pulse heterodyne lidar. As explained in the article, this function is required to correct the intensity of the signals recorded by the instrument (termed SNR as it is normalized to the level of white noise in the heterodyne lidar) from the variations of the instrument sensitivity with the range. This function is more complicated than with a direct-detection aerosol lidar as it is not given by the overlap between the laser beam and the telescope aperture, but has more to do with the efficiency of the heterodyne detection. However, an expression exists that predicts how it varies with the range as a function of two system parameters, namely the size of the telescope aperture D and the focal length f. The idea is to tune these two parameters until the corrected signal intensities match the attenuated backscatter measured by a nearby ceilometer. Of course, this method requires the presence of ceilometer nearby, but ceilometers are rather cheap instruments, and are deployed in great numbers in meteorological observation networks. The method opens the possibility to use a heterodyne wind lidar as an aerosol lidar and thus combine with a unique instrument the measurement of wind and aerosol backscatter profiles. This is of great interest for the characterization of pollution transport or the study of the atmospheric boundary layer.

There are several limitations to the method. One deals with the difference of the laser wavelength of a ceilometer (usually close to $0.9\mu m$) and a pulsed heterodyne lidar ($\lambda \approx 1.5\mu m$). The value of the attenuated backscatter are different at the two wavelengths, and the difference is dependant of the nature the aerosols. The method would thus be in trouble if several layers of different aerosols are present in the laser beam. This limitation is clearly discussed in the text. There is, however, a second limitation. The heterodyne efficiency does not depend solely of the instrument parameters, but also on the optical turbulence. This dependency appears in equation (2) of the article with the $\rho_0$ parameter. It is assumed in article that $\rho_0 >>$ D so that its effect on $A_e(R)$ can be neglected. This assumption is not justified. In practice, it can happen that the turbulence significantly degrades the heterodyne efficiency of the lidar. This is particularly the case when the beam is directed horizontally, a few meters above the ground, on a hot, sunny day. In the article, the vertical (or close to vertical) direction of the beam should alleviate the degradation as the optical turbulence drops very rapidly with the altitude, but it would be worth to have a short calculation of $\rho_0$ as a function of the range using a typical profile of $C_n{}^2$ and the formulation

of Frehlich and Kavaya for $\rho_0$ (equation 165).

This is a very good point. Discussion of the impact of refractive turbulence has been added as a new subsection 4.3 together with an additional figure (figure 7 in the new manuscript).

*So far we have neglected the potential impact of turbulence on $T_f(R)$ arising from the refractive turbulent parameter, $\rho_0$, in (2). An expression for $\rho_0$ is given in (Frehlich and Kavaya, 1991),*

$$\rho_0(R) = [Hk^2 \int_0^R C_n{}^2(z)(1 - z/R)^{5/3}dz]^{-3/5}, \tag{1}$$

*where $H = 2.914383$, $k = 2\pi/\lambda$, and $C_n{}^2(z)$ is the refractive turbulence at range z. We chose 3 profiles with constant $C_n{}^2(z)$, and a realistic vertical profile based on the most turbulent case presented by (Roadcap and Tracy, 2009). Figure 1 shows the impact that these different profiles have on $T_f(R)$, and the resulting re-sampling calculation of $\sigma_{T_f}(R)$ for two Doppler lidar instruments with different $T_f(R)$. Values of $C_n{}^2$ up to $10^{-14}m^{-2/3}$ have negligible impact on $T_f(R)$, and even the realistic profile only showed a slight increase in $\sigma_{T_f}(R)$ for the instrument with a focus set closer than infinity. This suggests that the impact of turbulence can be safely neglected for low values of $C_n{}^2$, and for most applications, can also be neglected when operating in the vertical. Hence, turbulence has no significant impact on the methodology described here for deriving the parameters f and D and their uncertainties from vertical profiles, but can be included for completeness.*

*However, it is clear that the turbulent impact should not be ignored when measuring at low elevation angles close to the horizon, where a profile similar to $C_n{}^2 = 10^{-13}m^{-2/3}$ may be possible. Fig. 1 shows that such a profile has a major impact on $T_f(R)$, especially in the far range. In these cases, the parameters f and D obtained from vertical measurements are still applicable, but $\rho_0(R)$ must also be calculated or estimated in order to derive the profile of attenuated backscatter, $\beta'(R)$.*

An additional paragraph has been added to the conclusion.

*The impact of turbulence on $T_f(R)$ was also investigated and was found to have no significant impact on the methodology described here for deriving the parameters f and D and their uncertainties from vertical profiles. However, the turbulent impact should not be ignored when measuring at low elevation angles close to the horizon, as it can modify $T_f(R)$ considerably, especially in the far range. In these cases, the parameters f and D obtained from vertical measurements are still applicable, but the turbulent contribution to $T_f(R)$ should included when deriving the attenuated backscatter coefficient.*

It is written in the abstract that the method proposed in the article is applicable to Halo Photonics heterodyne Doppler lidars. It is clear that it is tested on data from Halo Photonics lidars, but I do not see why it could not be ap-

[Figure]

Figure 1: Impact of turbulent parameter, $\rho_0$, on telescope focus function, $T_f(R)$ and relative uncertainties, $\sigma_{T_f}(R)$, for different $C_n{}^2$ profiles. a) selected profiles of $C_n{}^2$ with range; b) $T_f(R)$ and c) $\sigma_{T_f}(R)$ for Darwin, 21 June 2011 to 22 July 2012; d) $T_f(R)$ and e) $\sigma_{T_f}(R)$ for NSA 30 July 2014 to 31 December 2017.

plicable to heterodyne lidars from other manufacturers as long as they provide measurements of SNR. If that is true, it would be worth mentioning it in the abstract as it widens the applicability of the method.

*This is correct, and is now mentioned in the abstract and the conclusion.*

*Here, we present a methodology for deriving the telescope focus function using a co-located ceilometer for pulsed heterodyne Doppler lidars. The method was tested with Halo Photonics Streamline and Streamline XR Doppler lidars, but should also be applicable to other pulsed heterodyne Doppler lidar systems.*

*We have developed a method for deriving the telescope focus function and its uncertainty for pulsed heterodyne Doppler lidars, and applied the method to Halo Photonics Streamline and XR Doppler lidars.*

**1.2   Specific comments**

1. In the paragraph that follows equation (1), the meaning of the $\eta$ term is not explained.

   *$\eta$ is the detector quantum efficiency has been added to the text.*

2. Equation (5): the equation applies to relative uncertainties. This shall be made clear.

   *We have clarified this in the text.*

3. Line 28: why this -22.2dB SNR threshold?

   *The threshold is based on the SNR limit in Manninen et al.(2016) as the data we used has been processed with the same method. This threshold is based on the expected noise floor for the instruments considered here (Halo Streamline and Streamline XR) and should probably be modified for different instruments. A citation for the threshold has been added.*

4. Table 4 on page 13: the meaning of b in N(f-2, b) and N(f, b) is not explained in the legend.

   *The term b was a mistake, it is supposed to be D. This has now been corrected.*

---

## Author Response (AR2)

We have addressed all of the points raised by the editor (copied here and shown in black text), and include our responses to each point below (in blue text). Where there has been a major change in the manuscript we provide the original text (in black italics) and the new text (in blue italics).

**1 Editor**

I appreciate your clear answers to the reviewers remarks but I would suggest to answer some minor remarks from the reviewers directly in the manuscript (and not only in their answers). Could you please take into account the following corrections to be brought to the manuscript before publication:

1. p5, line 11 : as you included a new section about the impact of the turbulent parameter (rho). Could you just refer to section 4.2 to mention that it will be discussed later in the manuscript ?

   We have modified the manuscript as suggested:

   *We first assume that we can neglect $\rho_0$, and describe a method for estimating $f$ and $D$, together with their uncertainties, which can then be propagated to obtain the uncertainty in the attenuated backscatter coefficient. The impact of the $\rho_0$ parameter is discussed in section 4.3.*

2. p6, ligne3 : Could you please clarify why the interpolation is needed to follow reviewer 1, minor remark number 3.Instead of writing "when necessary" you could re-use the answer you made to the reviewer

   We have modified the manuscript as suggested:

   *Both ceilometer and Doppler lidar data were averaged to a common 30-minute, 30 m vertical resolution grid, using linear interpolation where necessary (only for one period from Darwin).*

   *Both ceilometer and Doppler lidar data were averaged to a common 30-minute, 30 m vertical resolution grid. If the Doppler lidar vertical resolution was larger than 30 m (as was the case for one period from Darwin), linear interpolation was used to match resolutions.*

3. p6, line3 : additionally to the reference for the SNR threshold used in the paper, could you also include one part of your answer that you did to reviewer 1 and 2 to explain the choice of this threshold.

   We have modified the manuscript as suggested and included an additional sentence:

   *The threshold is based on the expected noise floor for the instruments considered here (Halo Streamline and Streamline XR) and should probably be modified for different instruments.*

4. P10, legend of table 3 : can you clarify what is included in the total estimates to answer reviewer 1, major remark number 1 ? It could be something like : total estimates is the number of profiles without clouds, precipitation and multiple aerosol layer, good estimates is the number of good estimates after outlier removal using MAD.

   We have modified the table caption as suggested:

   *Total estimates refers to the number of ceilometer and Doppler lidar profiles suitable for comparison. Good estimates refers to the number of estimates remaining after outlier filtering with MAD.*

5. p12, figure4 a) : what does arbitrary unit means ?

   Uncalibrated lidar quantities are often plotted with units of $[a.u.]$ or arbitrary units, see for example the units for normalised signal given in this glossary for E-PROFILE:
   http://www.eumetnet.eu/wp-content/uploads/2016/10/ALC_glossary.pdf

6. P19, line 22: please correct "should included" by "should be included"

   We have modified the manuscript as suggested:

7. In line with reviewer 1, major remark number 3, could you mention somewhere in the manuscript that thanks to your method, after deriving the telescope focus function from a short-period with a co-located ceilometer, the Doppler lidar can be moved to another place not equipped with a co-located ceilometer but still able to provide both winds profils and attenuated backscatter profiles.

   We have included the following sentence in the conclusion:

[revised manuscript text omitted]